# Elaborating the Occurrence and Distribution of Per- and Polyfluoroalkyl Substances in Rivers and Sediment around a Typical Aging Landfill in China

**DOI:** 10.3390/toxics11100852

**Published:** 2023-10-11

**Authors:** Bingxu Quan, Jiawei Tang, Xiameng Niu, Peidong Su, Zhimin Zhang, Yitao Yang

**Affiliations:** 1School of Chemical & Environmental Engineering, China University of Mining & Technology (Beijing), Beijing 100083, China; quanbingxu@hotmail.com (B.Q.); spd1194042797@gmail.com (P.S.);; 2National Institute of Low Carbon and Clean Energy, Beijing 102211, China; 3School of Science, Tianjin University of Technology, Tianjin 300384, China

**Keywords:** perfluoroalkyl substances (PFASs), landfill, the suspended particulate matter–water partition coefficient (log K_d_), the hazard quotient method (HQ), ecological assessment

## Abstract

Per- and polyfluoroalkyl substances (PFASs) are bioaccumulative and widely distributed persistent organic pollutants (POPs). Understanding the distribution of and ecological risks posed by PFASs is critical, particularly for PFAS emissions and accumulation from a common urban pollution source. The transformation characteristics and ecological risks of PFASs from a typical aging municipal landfill leachate were systematically monitored and assessed over five years in this study. The results showed that the total PFAS concentrations (ΣPFASs) in the rivers were between 26.4 and 464.3 ng/L, whereas in sediment, ΣPFASs ranged from 9.5 to 58.5 ng/g (*w*/*w*). The presence of perfluorooctanoic acid (PFOA) was the most prominent PFAS in both water (39.4–152.3 ng/L) and sediment (1.1–56.1 ng/g). In a five-year monitoring study, the concentration of PFASs in the aging landfill decreased by 23.3%, with higher mean concentrations observed during summer (307.6 ng/L) compared to winter (250.4 ng/L). As for the pollution distribution, the suspended particulate matter–water partition coefficient (log K_d_) of carboxylic acid (PFCAs) and perfluoroalkane sulfonic acids (PFSAs) ranged from 1.53 to 2.65, and from 1.77 to 2.82, respectively. PFSAs and long-chain PFCAs exhibited a greater propensity for sediment association compared to short-chain PFCAs. An ecological risk assessment of four typical PFASs, PFOA, perfluorooctane sulfonate (PFOS), perfluorobutanoic acid (PFBA), and perfluorobutane sulfonic acid (PFBS), utilizing the hazard quotient method revealed that the rivers surrounding the typical aging landfill exhibited a low contamination risk for PFOA, while no ecological risks were associated with the other three FPASs. This study contributes to an enhanced comprehension of the occurrence, distribution, and risk of PFASs in the rivers in rivers and sediment surrounding a typical aging landfill site in China, thereby providing crucial reference information for ensuring water quality safety.

## 1. Introduction

Per- and polyfluoroalkyl substances (PFASs), as a diverse group of aliphatic compounds including multiple perfluoroalkyl moieties, is the summary term for a broad class of anthropogenic chemicals that have been used extensively over the past decades [1,2]. Centralized research has demonstrated that the PFASs are usually divided into carboxylic acids or sulfonic acids, which are called per-fluorinated carboxylates (PFCAs, C_n_F_2n+1_COOH) and perfluorinated sulfonates (PFSAs, C_n_F_2n+1_SO_3_^−^) based on different side moieties, respectively [3]. The high energy of C-F covalent bonds (116 kcal·mol-1) results in PFASs possessing chemically and thermally stable properties, making them widely useful in industrial and commercial applications. They can be found in many products such as firefighting foam objects, fast-food packaging, paper plates, fire-fighting foams, non-stick cookware [4,5] PFASs.

Perfluorooctanoate (PFOA) and perfluorooctane sulfonate (PFOS), as long-chain compounds (C ≥ 8), are two of the most studied and have been reported to have test records worldwide in a palpable range of environmental samples, including aqueous medium, soil, floating dust, wildlife, and even in human blood [6,7,8]. With continuous research reports, PFOS and its salts were officially listed on the POP list at the fourth meeting of the Stockholm Convention in May 2009, which led to restrictions and the suspension of related products, in which they were replaced by short-chain compounds dominated by perfluorobutane sulfonate (PFBS) [9,10]. Recently, however, more research has revealed that short-chain homologs like perfluorobutanoic acid (PFBA) and perfluorobutane sulfonate (PFBS) have become the dominant PFASs in the environmental medium. This indicates adverse environmental properties, in the form of a series of phenomena, that have occurred due to substituting long-chain PFASs and their respective precursors with short-chain homologues [11,12,13]. As a result, worldwide scholars are advocating for improved approaches to comprehensively assess the toxicity profiles and ecological risks of PFASs, due to their alternatives being structurally similar to the chemicals that they replace nowadays [14,15,16]. Numerous studies on PFASs pollutants have been directed toward the development of pollution characteristics and toxicity to aquatic organisms from typical areas [17,18]. As shown in Appendix A, the LC50 (50% of the lethal concentration) and EC50 (concentration that results in an effect in 50% of the sample population) were used to intuitively evaluate toxicity to organisms. However, it is worth noting that the fate and behavior of PFASs in differing or typical environments and conditions are mostly unknown, as are the effects of complex mixtures of compounds occurring at low doses but with continuing lifelong exposure [19]. To evaluate the toxicity and ecotoxicological risks posed by PFAS compounds to aquatic organisms in complex natural environments, the U.S. Environmental Protection Agency (EPA) established a comprehensive database of the criteria maximum concentrations (CMC) and criteria continuous concentrations (CCC) for PFOA and PFOS, specifically tailored for environmentally sensitive aquatic species [20,21]. Similar criteria have also been derived for PFASs in China through previous studies. While the majority of studies have primarily focused on natural rivers and lakes, there has been limited research conducted on water bodies in proximity to typical polluted sites.

A significant amount of daily care products, light-industrial-production wastewater, and packaging containing PFASs and their byproducts are increasingly being disposed of in municipal landfills [2]. PFASs are released into surface water, air, soil, and even groundwater after disposal, posing significant threats to the surrounding ecological environment [22]. Generally, landfill leachate was collected and treated by wastewater treatment plants (WWTPs) before their final disposal. However, these emerging contaminants (ECs) of PFASs can succeed in escaping from WWTPs, since they are not equipped to remove these classes of contaminants, and instead are the main sources of PFASs in aqueous solutions [23]. The concerns about the existence of PFASs surrounding ecological security and human life are getting serious, as previous epidemiologic conclusions of chronic exposure to PFASs are that they could cause neurotoxicity, hepatotoxicity, genetic defects, and behavioral disability [8]. Therefore, it is imperative and urgent to elaborate on the occurrence, distribution, and ecological risks associated with PFASs in the aquatic environment surrounding the typical-aged landfill.

Though there are multiple studies that have been published regarding the environmental occurrence and transformation of PFASs, few have focused on the changes in environmental and biological pollution caused by PFAS emissions from typical urban pollution sources on the time trends. Therefore, in this present study, the transformation characteristics of PFASs from a typical-aging municipal landfill leachate were systematically monitored and evaluated over 5 years, considering the spatio-temporal dimension. The main objectives of this study included the following: to (1) investigate the seasonal and annual existence and distribution of PFASs in leachate and surrounding rivers and sediments; (2) analyze and identify the pollution distribution using the suspended particulate matter–water partition coefficient (log K_d_); and (3) assess the ecological risk posed by four PFASs using the hazard quotient method and aquatic life criteria (ALCs) in aquatic environments.

## 2. Materials and Methods

### 2.1. Materials and Chemicals

The main chemicals used in the study included methanol (GR), methyl tert-butyl ether (MTBE, 99%), tetrabutylammonium hydrogen sulfate (TBA, 97%), ammonium acetate (chromatographic purity), glacial acetic acid (GR, >99.8%) and ammonia (GR, 50% *v*/*v*). All samples were detected for 13 PFASs purchased from the Sigma-Aldrich company, which included PFBA, PFBS, perfluorovalerate (PFPeA), perfluoroheptanoic acid (PFHpA), PFOA, PFOS perfluorononylic acid (PFNA), perfluorodecanoic acid (PFDA), perfluorotwelve acid (PFDoDA), perfluorohexanic acid (PFHxA), and perfluorohexyl sulfonic acid (PFHxS). In addition, necessary standards, such as quantitative mixed standard PFAC-MXB, internal standard ^13^C_4_-PFOS, internal standard ^13^C_4_-PFOA, internal standard ^13^C_4_-PFBA, and internal standard ^13^C_2_-PFDoDA were provided by Wellington Laboratories, Canada.

### 2.2. Target Landfill

Liulitun landfill is located in Yongfeng town, northwest of Haidian district, Beijing, with the approximate coordinates 40°02′–40°05′ N and 116°12′–116°14′ E. The pre-site area of Liulitun landfill covers 12 hectares and the total landfill area is 34.53 hectares after the first and second phases of the landfill expansion in 1999. More than 3 million people and 426 km^2^ area are served, with about 1000 t of garbage received per day. With the expansion and development of urbanization and the construction of new factories, the Liulitun landfill has gradually reduced the storage capacity of garbage sealed in recent years, resulting in the aging development of its leachate. Previous monitoring studies have proved that the UASB + A/O + MBR was the main technology for the advanced treatment of leachate, and exhibited high removal effects for the conventional pollutants (e.g., COD < 50 mg/L; NH_4_-N < 4 mg/L) to meet relevant standards, but still contained a high concentration of PFASs in leachate effluent [24]. Furthermore, the landfill is surrounded by residential areas as well as drinking-water-source protection areas (Jingmi diversion canal). Analysis of the pollutant occurrence and distribution, as well as risk assessment, is critical for environmental protection.

### 2.3. Sample Collection

A total of 12 sampling sites around the Liulitun landfill were set up for long-term sampling detection from March 2014 to September 2018. Based on the suspected water sample points of the Nansha River and the Jingmi diversion canal flow from west to east, and the tributaries between the two rivers that are distributed throughout the plant and living areas, from the south (average elevation 50–68 m) to the north (average elevation 45–50 m) until the inflow into the Nansha River (Figure 1), the collection area was divided into four sections, including the upstream discharge source (A-1), leachate discharge source and lateral diffusion zone (A-2), Nansha river estuary area (A-3), and upstream wetland park (A-4).

River and wetland lake-water samples were annually collected (except winter) at all 12 sampling sites, and a total of 336 (*n_w_*) water samples (surface layer, 0.25~0.45 m depth) were attained and numbered as W1 to W28. To reduce the error caused by rain and snow, we avoided collecting all samples within 5 days of rain. Sediment samples (*n_s_* = 88, surface to 10 cm deep) were also collected at 11 sampling sites (site 3 excluded) in the partial periods of March (S1) and July (S2) 2014, April (S3) and September (S4) 2015, May 2016 (S5), July 2017 (S6), June (S7) and October (S8) 2018, respectively. More detailed sampling information is given in Appendix A.

Water samples (>1 L) were collected with a stainless-steel water sampler before being stored in wide-mouthed PP bottles with screw caps, and the sediment samples were packed in self-sealing bags and brought back to the laboratory for refrigeration immediately.

### 2.4. Sample Preparation

#### 2.4.1. Drainage and River

Before enrichment and purification with 2 ng ^13^C_4_-PFOA and ^13^C_4_-PFOS standard samples, the supernatant of water samples was filtered through a 0.45 μm glass-fiber membrane to remove suspended particulates. WAX columella was then enriched and purified using 4 mL of 0.1% of NH_3_·H_2_O-CH_3_OH, 4 mL of CH_3_OH solution, and 8 mL of ultra-pure water. During sampling, the flow rate was kept at about 1 drop/second, and the WAX column was washed sequentially with 4 mL of 25 mmol/L acetate buffer and 4 mL of water. Following elution with methanol and nitrogen, the samples were vortexed and centrifuged before being analyzed.

#### 2.4.2. Sediment

The treated 0.2 g sediment sample was placed in a 15 mL polypropylene tube and treated with 2 ng ^13^C_4_-PFOA and 2 ng ^13^C_4_-PFOS, before vortexing and standing for 30 min. The temperature of the mixture was maintained at 50 °C, 10 mL of methanol solution was added, the solution was sonicated for 30 min, and finally it was shaken at 160 r/min for 16 h, then 10 mL supernatant was extracted after centrifugation at 4000 r/min for 10 min. Following that, another 5 mL of extract was prepared by adding 5 mL of methanol for secondary extraction at an oscillation speed of 250 r/min for 2 h and a 4000 r/min centrifugation speed for 10 min. The two extracts (15 mL) were combined and nitrogen-purged to 1.0 mL before being centrifuged at 4000 r/min for 10 min before being mixed with 39 mL of ultrapure water. The sample was subjected to solid-phase-extraction enrichment and concentration using a WAX SPE column before liquid chromatography.

### 2.5. Instrumental Analysis and Quality Assurance

Instrumental analysis was employed using high-performance liquid chromatography (HPLC) with a dual pump and autosampler (Ultimate 3000, Dionex, Sunnyvale, CA, USA), coupled to electrospray ionization tandem mass spectrometry (ESI-MS/MS, API 3200, Allen-Bradley, Milwaukee, WI, USA). The column of C18 reversed-phase chromatography was selected for analysis, which was washed by using methanol and 50 mM acetic acid as the mobile phase and gradient elution. Specific chromatographic conditions included the fact that the chromatography column was Acclaim 120 C18 (4.6 × 150 mm, 5 m), the mobile phase A was methanol, the mobile phase B was 50 mM ammonium acetate solution, and the sample volume was 10 μL, with a flow rate of 1 mL/min. Furthermore, the optimized mass spectrometry conditions were as follows: 0.24 MPa of air curtain gas, 0.021 MPa of collision gas, and 2000 V of ion spray voltage at 400 °C.

All laboratory consumables and dissection tools were checked for contamination, and some blank samples were collected for every ten samples. The two internal standards (2 ng of 13C4-PFOA and 13C4-PFOS) mixed with other standard solutions were added to the samples before extraction. The recoveries and reproducibility of PFASs in samples were calculated, and are summarized in Appendix A. The limit of detection (LOD) was defined as the concentration that yielded a signal-to-noise (S/N) ratio of 3, while the limit of quantification (LOQ) was defined as the concentration that yielded an S/N ratio of 10. Under 2000 ng/L standard addition, the recoveries of PFASs in actual water and sediment were 90–208% and 102–159%, while the relative standard deviations (RSD) were between 1–8% and 5–43%, respectively.

### 2.6. Data Analysis

The analytical results lower than LOD were marked as n. d. (not detected) and zero was assigned for statistical analysis. The propensity of a PFAS’s adsorption distribution behavior to the aqueous phase or sediment has great significance in us understanding the migration, transformation, and fate of PFASs in the environment. The purpose of introducing the standardized partition coefficient (*K_d_*) is to explore the solid–water phase partition coefficient of PFASs.
(1)$$K_{d}{=C’}_{\mathrm{sediment}}{/C}_{\mathrm{wt}}\;\times \;1000$$
where *C′_sediment_* (ng·g^−1^ dw) is the concentration level of PFASs in sediments, while *C_w_* is the concentration of PFASs in the aqueous phase.

To assess ecological risk, the hazard quotient (*HQ*) method was used. The *HQ* was defined as the measured exposure concentration (*MEC*) of a pollutant divided by the predicted non-effect concentration (*PNEC*). The *HQ* was calculated using Equation (2). In this study, the ecological risk assessment was conducted for PFBA, PFOA, PFBS, and PFOS. PNECs were derived from previously published research [25,26,27].
(2)$$\mathrm{HQ}=\frac{{\mathrm{MEC}}_{\mathrm{water}}}{{\mathrm{PNEC}}_{\mathrm{water}}}\;\mathrm{or}\;\frac{{\mathrm{MEC}}_{\mathrm{sediment}}}{{\mathrm{PNEC}}_{\mathrm{sediment}}}$$

A classification of *HQ* values is shown below, which was divided into four levels to indicate the level of risk presented. The *HQ* categories were

(1)*HQ* < 0.1, no risk;(2)0.1 ≤ *HQ* < 1.0, low risk;(3)1.0 ≤ *HQ* < 10, moderate risk;(4)*HQ* > 10, high risk.

## 3. Results and Discussion

### 3.1. Concentration and Distribution of PFASs in River and Sediment

#### 3.1.1. Spatial Trends and Composition of PFASs

The concentrations of analytes indicated obvious differences between the locations and sampling dates, reflecting the composition of emission sources, impact of seasonal changes, and partitioning behavior of PFASs. For the landfill drainage (site 3) and surrounding surface rivers, the average total concentrations of the 11 PFASs (Σ_11_PFASs) in the aqueous phase are shown in Appendix A. The maximum concentration of PFASs (2235 ng/L in W4, August 2014), detected in this study exceeded the maximum concentrations reported in landfills in Scandinavia (27–697 ng/L) and Norway (590–757 ng/L), and was lower than in Australia (73–25000 ng/L), Germany (137–2509 ng/L) and the United States (270–2500 ng/L), indicating the concentration of PFASs in this effluent was of a medium level [28,29,30,31]. Furthermore, among the 11 target sample sites from the surface rivers/polluted streams, in addition to the highest concentration at sample site 4 (464.3 ng/L) which is near the effluent discharge area, the remaining 10 sample sites of Σ_11_PFASs ranged from 26.4 to 297.2 ng/L. The PFAS concentrations in this study were comparable to the main river basins (0.05–263 ng/L) in China [27,32,33,34], and also comparable to or lower than rivers in Korea (1.17–40.63 ng/L) [35], South Africa (0–38.5 ng/L) [36], USA (0–65.5 ng/L) [37], Sweden (1–60 ng/L) [38], and Japan (0–431.4 ng/L) [39]. Typically, the distance from the effluent source and the direction of the water flow also had a significant effect on the results, i.e., the Σ_11_PFASs of sample site 2 (77.6 ng/L), 8 (120.4 ng/L), 7 (154 ng/L) rapidly decreased with the direction of the stream. Data from sample site 11 (72.6 ng/L) in the upper reaches of Nansha River and downstream site 10 (102.7 ng/L) indicated that the tributaries of the drainage had increased the concentration of PFASs. Without considering the influence of human activities, it is speculated that water self-purification and sediment adsorption may play a positive role in reducing PFASs. On the contrary, the accumulation of Σ_11_PFASs in sediment (Appendix A) showed different phenomena. Among 11 sites, the highest Σ_11_PFASs of 58.8 ng/g in sediment was detected at site 4, followed by 32.2 ng/g at site 9, 31.1 ng/g at site 5, and 23.4 ng/g at site 10. Sample site 12 from the wetland park was used as the control group, and the Σ_11_PFASs in the sediment were still 12.6 ng/g, similar to sites 6 (20.5 ng/g), 7 (18.3 ng/g), and 8 (19.4 ng/g) which could be attributed to the fact that the water in the wetland park is stagnant water and pollutants have been deposited here for years. Another reason could be explained by the fact that the tributaries where sites 6, 7, and 8 are located are shallow (0.5 ~ 2 m), and thus, most of the pollutants are transferred by the stream downstream.

The concentrations of each PFAS in the water samples are summarized in Figure 2. Among the 11 target PFASs in the 11 sites (except site 3), 6 compounds out of the 11 PFASs measured in rivers, PFHxA, PFHpA, PFNA, PFDA, PFDoDA, and PFBS, were not detected in some periods and locations, as their detectable rates were 78.2, 64.3, 74.7, 37.3, 54.8, and 83.7%, respectively. The predominant PFAS in site 3 was PFOA, which accounted for 41% of Σ_11_PFASs, whereas the dominant PFASs near the discharge source (site 4) were PFOA, PFBA, PFPeA, PFHpA, and PFBS, with the concentrations ranging (means) from 39.4 to 152.3 (96.5), 31.5 to 175.8 (72.8), 20.3 to 144.6 (58.9), 35.4 to 78.6 (48.7) and 16.2 to 65.7 (37.8) ng/L, respectively, indicating that PFASs were ubiquitous in the surface river phase. Additionally, the concentration levels of PFSA sampling in other sites were lower than those of PFCAs, suggesting that the distribution of PFAS components in the surrounding streams was similar to that in the effluent, and PFSAs were the dominant component. For the composition of PFASs in the sediment, in addition to the low concentration at the sampling sites 1 and 2, with the PFHxS, PFBS, and PFOA as the main components; the essential component in the remaining sampling points was PFOA, accounting for more than 55% of the PFAS concentrations (as shown in Appendix A). Note that the PFAS concentration of sampling site 4 was relatively high (58.8 ng/g) due to its closeness to the drainage outlet, which also led to the accumulation of PFOA as the dominant component in sampling sites 5 and 9 downstream.

#### 3.1.2. Five-Year Temporal and Seasonal Distribution

In a five-year tracking detection, we observed the temporal distribution of PFASs collected from water and sediment samples between 2014 and 2018. For the water samples, significant variation in the PFAS levels according to leachate effluent was found, compared to a slight fluctuating decrease in PFAS concentrations at other sampling points. As presented in Appendix A, the secondary effluent concentration of PFASs at sampling site 3 decreased by 23.3% in five years, i.e., from 2097 to 1607 ng/L, followed by a reduction rate of 11.6, 23.5, 53.0 and 32.7% at sampling sites 4, 5, 9 and 10. The concentration changes of the above sampling sites indicated that the emission source played a leading role in the downstream water solution. The PFAS concentration variation of sampling sites 6, 7, and 8 showed a wave-like change with mean values (median, and standard deviation) of 192.4 (192.1, 37.9), 154.1 (159.0, 23.7) and 120.4 (131.8, 29.9) ng/L, respectively. As it did not show a correlation with leachate drainage, it was presumed to be mainly affected by human activities in the surrounding active area. In addition, on the samples from all sites over the five years, it was found that the mean concentration of PFASs in summer (June–September) (307.6 ng/L) was higher than that in winter (November–February) (250.4 ng/L). According to the research of Gallen et al. [7], the concentration of PFASs in leachate is affected by many factors, including the dilution effect and lag effect caused by a change in climate conditions, precipitation, temperature and the amount of treated wastewater, which could have a significant impact on the concentration, as well as the important impact from human activities.

The five-year temporal distribution of PFASs in surface sediments is shown in Appendix A. Among the 88 sediment samples, there was a high concentration of PFASs at site 4 near the effluent of the landfill, with a mean (standard deviation) value of 57.2 (6.45) ng/L. The concentrations of PFASs were also high in sediment samples from sites 5, 9 and 10, which was similar to the trend of water samples, and the higher concentrations were at S3, S5, and S7, corresponding to April 2015, May 2016, and June 2018, respectively. For the temporal and seasonal distribution of other sample sites, we did not observe significant seasonal differences, which suggests that the sediment was less affected by the concentration variation of PFASs in the water.

### 3.2. Partitioning of PFASs between the Water Phase and Solid Phase

Due to the existence of hydrophobic and hydrophilic groups in PFASs, a series of migration and transformation processes will occur after them entering the environment. In the study of Ding et al. [5], the adsorption and distribution behavior of PFASs in water and solid phases (surface sediments) were studied using the sediment–water partition coefficient (K_d_) and the occupancy in sediments (*φ*, the mass concentration of PFASs adsorbed in sediment to the total concentration of those in the water and sediment phase). The concentrations of PFASs in the water-dissolved phase and solid phase are presented in Appendix A. It can be observed that the log K_d_ value of PFASs ranged from 2.00 to 2.67, and the average log K_d_ value (2.11) of area 2 (site 4–8) was significantly less than those from area 1 (sites 1 and 2) (2.42) and area 3 (site 9–11) (2.40). This was attributed to the variances that existed for the concentration of PFASs between the water phase and the sediment; e.g., the average PFAS concentration in the sediment of A-2 area was 29.9 ng/g, and 245 ng/L in the corresponding aqueous solution, while the PFAS concentrations in the sediment (water solution) of the A-1 and A-3 areas were 11.3 (61 ng/L) and 25.8 (108.3 ng/L) ng/g, respectively. The appearance of these differences depended on the physical, chemical, and biological conditions of the water body and varies within and among aquatic ecosystems. More importantly, the variance can be related to the nonequilibrium status of individual PFASs between the sediment phase and water-dissolved phase. Hence, a total of 6 typical sampling sites of the stream (sites 1, 4, 5, and 9), side branches (site 7), and Nansha River (site 10), near where the discharge outlet was located, were selected for analysis. As shown in Figure 3, the log K_d_ of PFCAs and PFSAs ranged from 1.53 to 2.65, and 1.77 to 2.82, respectively. From the log K_d_ values, it could be seen that PFSAs and long-chain PFCAs were more inclined to associate with sediment than short-chain PFCAs at every site. PFOA and PFOS were the predominant PFASs in the sediments, with a mean average contribution of 21.41%, and 28.16%, respectively. This result agrees with previously reported research, and could be explained by the stronger potential of long-chain PFASs to interact with the solid phase [16,38]. It can be also seen in the contribution of PFASs bound to sediments to the total concentrations of PFASs (φ). The values of short-chain PFCAs, including PFBA, PFPeA, and PFHpA, were generally less than 10%, except for site 9, while the values of long-chain PFASs, such as PFOA and PFOS, were more than twice as high as those. In general, the carbon-chain lengths of PFASs determined their partitions in the sediment and aqueous phases, which could lead to their long-range transportation with currents, sedimentation, or accumulation in the sediment. This also further revealed that the reason for the higher log K_d_ value of area 2 was attributed to the accumulation of predominant long-chain PFASs, including PFOA and PFOS, in areas far from the leachate PFAS source in area 2. Moreover, it also revealed that the sediment-bound fraction is a key estimated factor that can determine the results of the assessment of the environmental load of PFASs. Therefore, greater attention should to given to the solid-bound fraction compared to the water-dissolved fraction to better understand the environmental behavior and fate of these PFASs.

### 3.3. Risk and Evolution Assessment of PFASs

In this study, to determine ecological risks to aquatic organisms in rivers around a typical aging landfill under PFAS exposure, a preliminary environmental risk assessment using the HQ method was conducted for PFBA, PFOA, PFBA, and PFOS, four of the most frequently detected and of-public-concern PFASs in water environments. PNECwater and PNECsediment of these typical PFASs were obtained from previous studies. The PNEC values of PFBA, PFOA, PFBA, and PFOS for the water were 21, 0.57, 72, and 1 × 10^−3^ mg/L, respectively; the PNEC values of PFBA, PFOA, PFBA, and PFOS for the sediment were 8.3, 0.19, 37.08 and 0.067 mg/kg (*w*/*w*), respectively [25,26,27,40]. The HQ values for PFBA, PFOA, PFBS, and PFOS in the water and sediment were calculated for all sites in rivers around the aging landfill, and the results are presented in Appendix A. The results indicate that water samples collected from all sampling sites in the present study did not contain concentrations of PFBA, PFOA, PFBS, and PFOS likely to pose an ecological risk (HQ < 0.1). Furthermore, all sediment samples collected at all sampling sites showed acceptable concentrations of PFBA, PFOA, PFBS, and PFOS, except for the POFA at sampling point 4, which was located close to the landfill and had an HQ value of 0.25 (Figure 4). Moreover, the closer to the landfill the site was, the higher the HQ value. Compared to other rivers, the PFOA concentrations (1.5–48.05 ng/g) in the sediment in this study were higher than those measured in Songhua River (0.1–0.87 ng/g) [41,42], Liao River (0.08–0.49 ng/g) [43,44], Yellow River (3.23–7.8 ng/g), Yangtze River (007–0.55 ng/g) [45], and Pearl River (0–0.15 ng/g) [46,47]. The calculated HQ values in the sediment for the above river basins, calculated using Equation (2), were lower than those in this study, and pose a low risk or no ecological risk.

According to the results in Section 3.2 and those of previous research reports [1,31,45,48], PFASs were more easily absorbed into sediments with an increase in carbon-chain length. This may explain why PFOA was found in low-risk levels in the sediments in rivers around the aging landfill. Though the ban and/or restrictions on the use of PFOS and PFOA have contributed to the level decrease in PFOA in the environment in recent years, China’s full implementation of the ban was late and many industries are still highly dependent on PFASs, which leads to persistent ecological risks of PFOA. Furthermore, previous studies of the ecological risk of PFOA have mainly been focused on surface water, and most stated the risk to be low or at an acceptable level. They neglected the critical role of sediment as an internal pollutant, which can not only accept pollutants but can also release them into water bodies again. In addition, due to the refractory degradation properties of PFASs, the risk of PFOA in sediments appears to persist for a long time [49]. Therefore, to ensure the ecological security of the water environment, it is necessary to monitor PFOA in water and sediments in various rivers around the aging landfill.

## 4. Conclusions

In this study, a total of 11 PFASs were found in both the surface water and the sediment in the rivers around a typical aging landfill in China. The total PFAS concentrations (ΣPFASs) in the rivers were between 26.4 and 464.3 ng/L. The predominant PFAS in the discharge source was PFOA, which accounted for 41% of Σ_11_PFASs (1750.7 ng/L), whereas the dominant PFASs near the discharge source were PFOA (39.4–152.3 ng/L), PFBA (31.5–175.8 ng/L), PFPeA (20.3–144.6 ng/L), PFHpA (35.4–78.6 ng/L), and PFBS (16.2–65.7 ng/L). ΣPFASs in sediment were between 9.5 and 58.8 ng/g (*w*/*w*). The predominant species in the sediment was PFOA (1.1–56.1 ng/g). It was found that the PFAS concentration at the aging landfill decreased by 23.3% in five years, and the mean concentration of PFASs in the summer (June–September) (307.6 ng/L) was higher than that in the winter (November–February) (250.4 ng/L). Regarding PFAS concentrations in the sediment, they were less affected by the concentration variation of PFASs in the water. As for the sediment–water partition behavior of PFASs, results showed that the PFSAs and long-chain PFCAs were more inclined to associate with sediment than short-chain PFCAs. PFOA and PFOS were the predominant PFASs in the sediments, with a mean average contribution of 21.41%, and 28.16%, respectively. The ecological risk assessment for PFBA, PFOA, PFBS, and PFOS showed that only PFOA in the sediment posed a low risk in the rivers around the aging landfill. It is imperative to conduct the systematic and periodical monitoring of PFOA levels in both water and sediment across diverse river systems adjacent to the aging landfill. By conducting this study, we were able to gain a greater understanding of the presence and risk of PFASs in rivers and sediment around typical aging landfills in China.

## Figures and Tables

**Figure 1 toxics-11-00852-f001:**
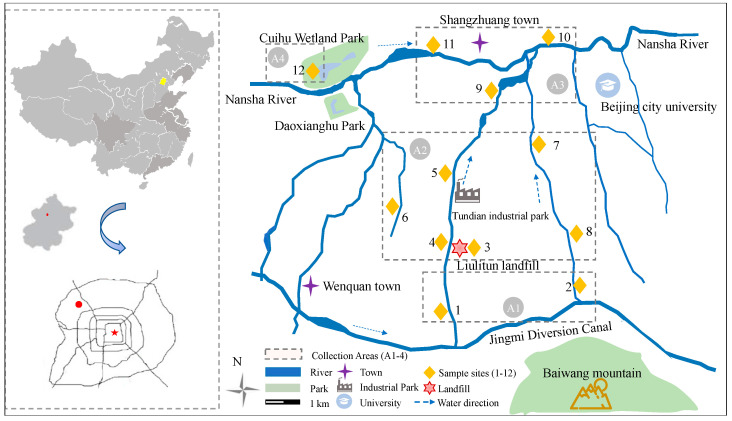
Samples sites around Beijing Liulitun Landfill.

**Figure 2 toxics-11-00852-f002:**
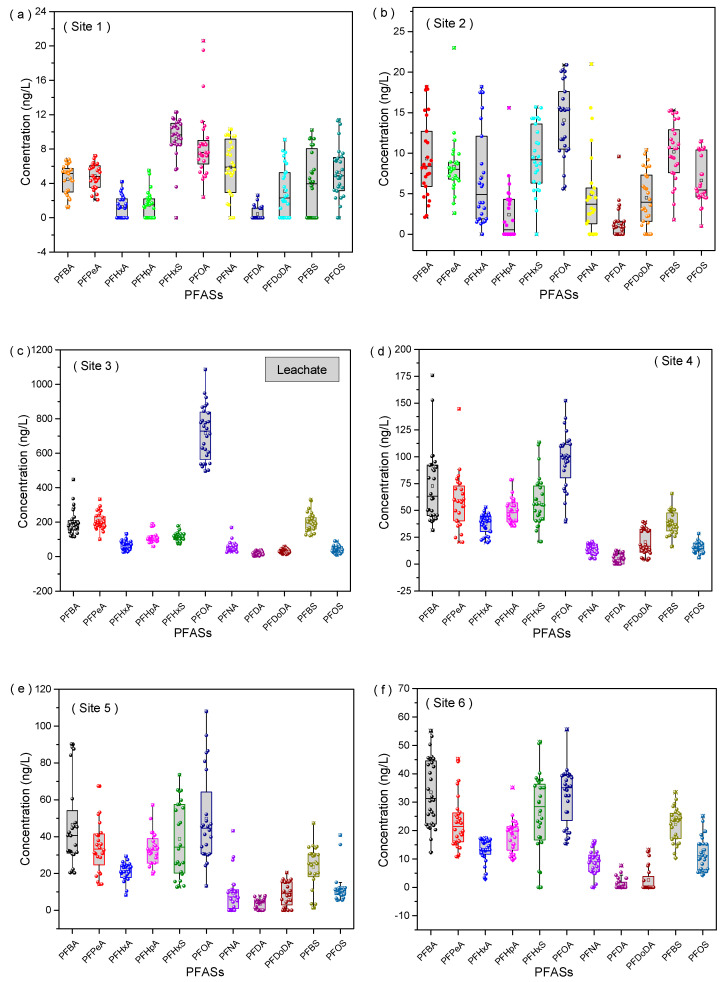
Box-and-whisker plots of PFASs in aqueous solution from 12 sampling sites around the target landfill in five years. Data with <80% detected concentrations in all sites were excluded.

**Figure 3 toxics-11-00852-f003:**
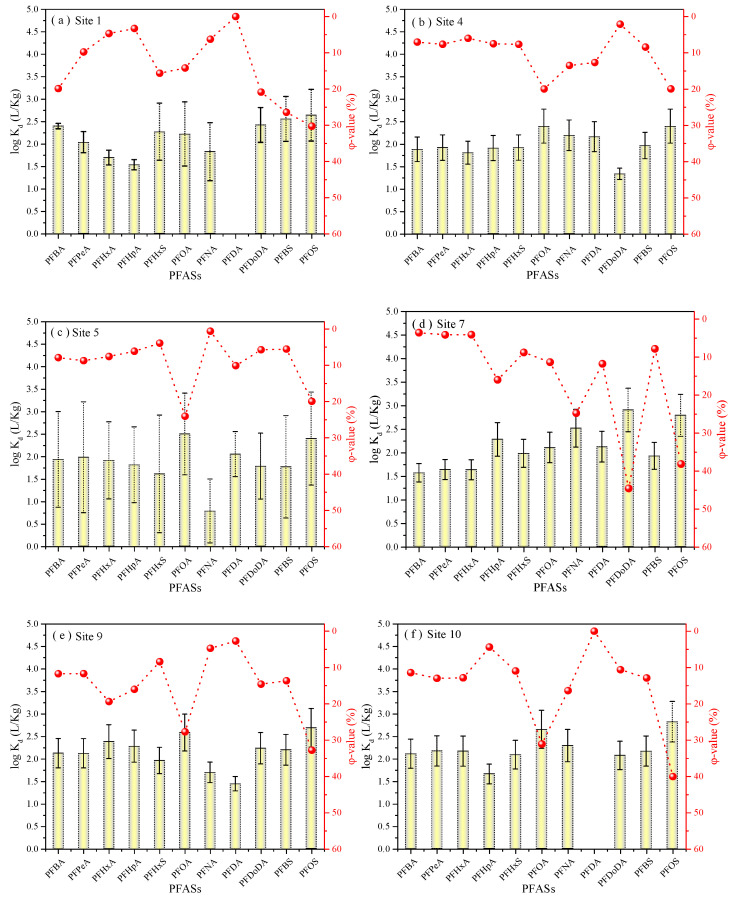
Average log K_d_ values and *φ* values of individual PFASs from sediment and river (stream) in typical areas (sites 1, 4, 5, 7, 9, and 10) of sampling sites 1 (**a**), 4 (**b**), 5 (**c**), 7 (**d**), 9 (**e**), and 10 (**f**).

**Figure 4 toxics-11-00852-f004:**
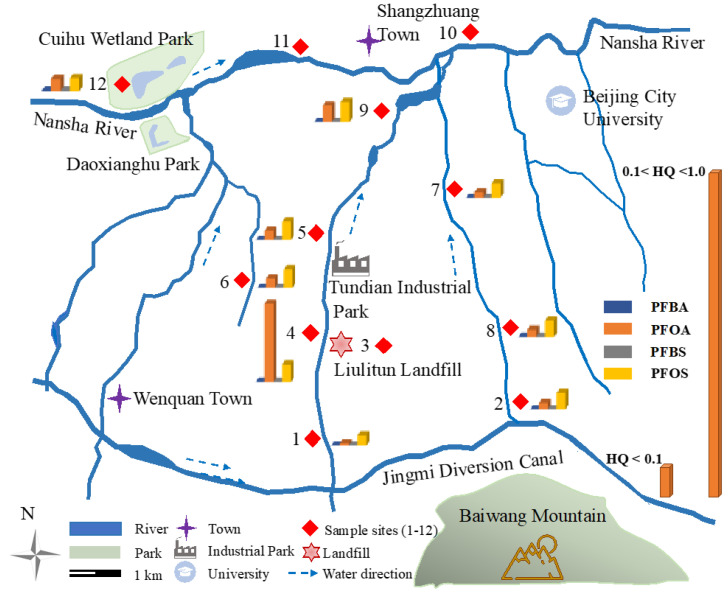
Hazard quotient values (HQs) for the PFBA, PFOA, PFBS, and PFOS in the sediment of the rivers around the aging landfill.

## Data Availability

The data involved in this study have been presented in the paper.

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
