# Peer review of "Elaborating the Occurrence and Distribution of Per- and Polyfluoroalkyl Substances in Rivers and Sediment around a Typical Aging Landfill in China"

_toxics, 2023, doi:10.3390/toxics11100852_

Round 1
Reviewer 1 Report
Review. Manuscript ID: toxics-2611215
The manuscript presents interesting results of monitoring of such organic micropollutants as polyfluoroalkyl substances. As a rule, modern purification facilities do not purify emissions into the atmosphere and industrial wastewaters from polyfluoroalkyl substances, and they (pollutants) income into the environment and accumulate there. Earth surface sites, where the household wastes polygons are situated, are also sources of polyfluoroalkyl substances, from which pollutants income into water bodies with drainage waters. The manuscript presents concentrations of polyfluoroalkyl substances and of products of their transformation in water and bottom sediments of rivers, which flow in the area of household wastes polygons.
On the base of the assessment concentrations of pollutants distribution between water phase and bottom sediments in rivers beds, recommendations for control of pollutants of this class in aquatic ecosystems are proposed. Correlational analysis of concentrations of polyfluoroalkyl substances and of products of their transformation in the studied samples would considerably increase the scientific level of the research done. Evidently, while assessing the environmental risk, the authors should have discussed main biota species inhabiting the studied water bodies and select corresponding values of non-effect concentration (PENC). Such an approach would assess ecological risk for biota in the studied water bodies from presence of polyfluoroalkyl substances and their transformation products there at the detected concentrations level.
It is necessary to pay attention to presented results of the assessment of accuracy of pollutants determination, lines 187-197. There can be an error here?
Measurements results have to be presented by valuable figures, e.g., Germany (137-2500 ng/L) rather than Germany (137-2509 ng/L), lineа 221 and further in the text.
Reviewer 2 Report
In the manuscript, the authors investigated the occurrence and distribution of PFASs in the rivers and sediment. This work reported the concentration of the 11 detected PFASs, with PFOA as the predominant compound. The experiment design is reasonable, and the data can support the statement. It is acceptable subject to a minor revision.
Line 94, double check the name the TBA, an IUPAC name is suggested.
Line 183, the format of the chemical names are incorrect.
Line 202, typo, the abbreviation should be “PNEC”
Line 187, the unit should be consistent, ie., convert the ppb to the ng/L as in other section of the manuscript.
Reviewer 3 Report
Comments and recommendations to the authors
In the Introduction, the data presented on toxicity to animal and human organisms are rather scarce (lines 57-67). More examples should be added, which could be more clearly presented by including them in a table.
In the Material and the methods it is specified about the collection of crucian carps from rivers (lines 137-141) and processing them (lines 145-147) for the same monitoring period, but in the Results, including the supplementary material, there are no results and discussions about the crucian carps analysis.
In the Results:
- In Figure S1, the spatial distribution of the mean PFAS concentration at each sampling site is not clear (the concentration at station 3 appears lower than in the graph).
- Figure 4 needs to be redone, as it is overloaded with too much overlapping data and the differences between the investigated areas cannot be well visualized.
- In subchapter 3.3 toxic effects on the sampled crucian carps, organisms that may have a role in the ecosystem in transferring these contaminants to the food chain, including humans, were not presented. The work is incomplete in this respect.
-
Reviewer 4 Report
The authors investigated the occurrence and distribution of per- and polyfluoroalkyl substances in rivers and sediments around an aging landfill in China. The research was conducted over a period of 5 years. The topic taken up by the authors is original and the data obtained are interesting.
The manuscript requires a minor correction. Detailed comments:
1. The utility of this study could be more clearly highlighted in the manuscript.
2. Introduction - briefly explain the motivation for undertaking this research, its relevance and originality, where it fits into the development of the field, and why it should be of interest to Toxics readers.
3. For readers to quickly catch your contribution, it would be better to highlight major difficulties and challenges, and your original achievements to overcome them, in a clearer way in introduction.
Round 2
Reviewer 3 Report
General remarks:
Both in the abstract of the paper and in the conclusions the existence of the environmental risk is not well formulated and is treated superficially. The wording needs to be much more precise, as there is sufficient evidence in the content of the paper.
Reply to the authors at point 2: These results should be processed, compared with other similar studies and included in the paper. Maybe they will have an impact in the formulation of conclusions, especially as the monitoring in this study does not provide current data on pollution in this area!
Reply to the authors at point 5: There could be a risk in the area and people end up eating contaminated fish. We need to prevent this by all means, in particular by rigorously investigating contaminated fish. That's why I think that the results obtained from the fish analysis should be included in the manuscript.

Round 3
Reviewer 3 Report
Comments to authors
The authors' reply managed to explain all the ambiguities and took into account the recommendations on completing the abstract and conclusions.
